# Towards a gold-standard of HER2 breast cancer biopsies using supervised learning based on multiple pathologist annotations

**Editor:**

## Abstract

Breast cancer is one of the most common cancer in women around the world. For diagnosis, pathologists evaluate biomarkers such as HER2 protein using immunohistochemistry over tissue extracted by a biopsy. Through microscopic inspection, this assessment estimates the intensity and integrity of the membrane cells' staining and scores the sample as 0, 1+, 2+, or 3+: a subjective decision that depends on the interpretation of the pathologist. This paper presents the preliminary data analysis of the annotations of three pathologists over the same set of samples obtained using 20x magnification and including $1,252$ non-overlapping biopsy patches. We evaluate the intra- and inter-expert variability achieving substantial and moderate agreement, respectively, according to Fleiss' Kappa coefficient, as a previous stage towards a generation of a HER2 breast cancer biopsy gold-standard using supervised learning from multiple pathologist annotations.

**Keywords:** Breast cancer, HER2 score, intra-expert variability, inter-expert variability, biopsy score consensus.

## 1. Introduction

According to the World Health Organization (21), in 2020, 2.3 million women worldwide were diagnosed with breast cancer, and $685,000$ died from this disease. For breast cancer detection using HER2 biopsies, immunohistochemistry (IHC) or fluorescence in situ hybridization (FISH) analysis are available. IHC is fast and technically more effortless and cheaper than FISH (18); however, the evaluation is subjective and shows intra- and inter-expert variability (17). In contrast, FISH gives quantitative results with less intra- and inter-observer variability. Nevertheless, FISH is a more time-consuming method, requires expensive reagents, fluorescence microscopy equipment, among others.

Typically, HER2 scoring is performed manually by microscopic examination, estimating the intensity and completeness of membrane cell staining and scoring the sample as one of four labels: 0, 1+, 2+, and 3+; where 0 and 1+ are negative, 2+ is equivocal, and 3+ is positive (20). In this sense, HER2 scoring is based on a subjective decision that depends on the pathologist's experience and interpretation (1; 7; 14). This non-objective decision could result in different diagnoses reached by different pathologists (inter-pathologist variability). Moreover, the same HER2 sample evaluated by the same pathologist at a different time could lead to different diagnoses (intra-pathologist variability) (6). There is evidence of inter-pathologist variability of up to approximately 7.7% (12). In this scenario, reproducibility of HER2 scoring is a challenging task.

In recent years, there have been research works aimed at automating the HER2 scoring for breast cancer (13; 2; 3). However, despite such works, it has not yet been massively implemented in the clinical setting, mainly due to the lack of reliable data that would guarantee a robust evaluation of the proposed methods. In addition, there are still no standard ways to compare the results obtained with different methods. The correlation of IHC with FISH was used to compare expert versus automated HER2 assessment (8). The use of concordance analysis is a different approach to performance assessment in the absence of ground truth. A valid alternative is to ask domain experts for their opinion on specific cases to generate a gold standard (10). Published algorithms for classifying breast cancer biopsies are usually evaluated based on their correlation with expert-generated classifications. However, even when there has been an advance in having public datasets, scores are based on the subjective opinion of only one expert, in most cases.

Motivated by this challenge, this research aims to achieve, as the ultimate goal, the consensus opinion of various expert pathologists on HER2 breast cancer biopsy cases, using supervised learning methods based on multiple experts. In this sense, we aim to generate a public breast cancer gold-standard, combining pathologists' opinions correlated with FISH results, to be used as a training/testing dataset in future machine learning methods for automatic HER2 scoring. In this article, we present the methodologies for the collection of samples and expert's opinions. In addition, as preliminary results, we assess intra- and inter-expert variability using the manual score given by three expert pathologists.

This article is organized as follows. Section 2 reviews research work in the area, justifying the need for a gold-standard for HER2 scoring. The section 3 is devoted to describing in detail the process of collecting biopsy sections and expert opinions, as well as giving an overview of methods for combining expert opinion. Our preliminary data analysis results are presented in section 4. Conclusions are found in section 5.

## 2. Related Work

The importance of having an image dataset containing absolute truth labels has been well demonstrated in many computer vision applications: handwriting recognition (15), face recognition (19), and indoor/outdoor scene classification (16), among others. However, a ground truth represents the absolute truth for a given application that is not always available. Unfortunately, for many applications, especially in biomedicine, it is impossible to have absolute truth, and a valid alternative is to ask the opinion of experts in the field on specific cases to generate a gold-standard (10). The need for a gold standard in biomedical applications has been demonstrated in Pap smear classification (11), human sperm segmentation (5), and classification of subcellular structures (4), among others.

There are no publicly available gold standards for HER2 scoring that combines various pathologist' opinions. Instead, several research groups have independently collected images of breast cancer biopsies and performed different experiments with different evaluation metrics. A list of several breast cancer biopsy datasets currently used in publications on the automatic HER2 scoring is shown in Table 1. Only one of them is a public dataset (17).

| Publication | Cases | Experts | Source |
| --- | --- | --- | --- |
| Qaiser et al.(17) | 86 | 2 | Nottingham University Hospitals, NHS Trust, Nottingham, UK |
| Khameneh et al.(13) | 48 | 1 | Acıbadem Maslak Hospital, Istambul, Turkey |
| La Barbera et al.(3) | 360 | 1 | IPATIMUP Diagnostics, University of Porto, Portugal |
| Anand et al.(2) | 52 | 1 | Nottingham University Hospitals, NHS Trust, Nottingham, UK |

Table 1: **Summary of previous datasets for automatic HER2 scoring.**

## 3. Material and Methods

### 3.1 Data collection

#### 3.1.1 HER2 breast cancer biopsies

The dataset comprised whole slide images (WSI) extracted from 30 cases of invasive breast carcinomas. The Biobank X managed the collection of HER2-stained slides obtained from two pathology laboratories: (1) Anatomic Pathology Service of Hospital Y and (2) Anatomic Pathology Service of Hospital Z.

All the biopsies have known positive and negative histopathological diagnoses (equally distributed in categories 0, 1+, 2+, and 3+). In addition, all samples were subjected to supplemental FISH analysis, according to the ASCO/CAP guidelines (20). Each of these samples was digitalized using a WSI tissue scanner (Hamamatsu NanoZoomer). Over each WSI, the tumor regions were marked by an expert pathologist as regions-of-interest (ROIs), see Figure 1 (left). There were considered between $3-4$ ROIs in each sample.

Then, to simulate actual microscopic examination performed by pathologists and according to their opinion, magnification of $20x$ was used to crop non-overlapping rectangular sections, approximately $15-17$ in each ROI. A total of $1,252$ biopsy patches were obtained. Each one of these patches was geometrically transformed (rotation, vertical flip, and horizontal flip), looking to evaluate intra-expert variability. With all biopsy patches transformed two times, the complete dataset has $3,756$ images.

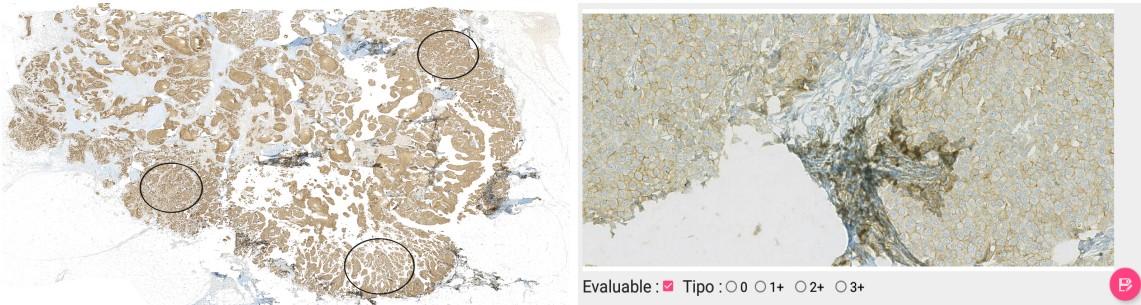

Figure 1: **Left:** Whole-slide-image with the regions-of-interest (ROIs) marked on by an expert pathologist. **Right:** Screen-shot of the Android application interface for collecting pathologist's opinions.

### 3.1.2 Biopsy scores from pathologists

An Android application was specially designed and developed to collect the expert pathologists' opinions. In this sense, each pathologist had the same device under the same conditions to have a controlled scenario to evaluate inter-observer variability.

The biopsy patches were presented randomly, and for each image, the pathologist indicated whether the image was evaluable or not and assigned a score among 0, 1+, 2+, and 3+ (see Figure 1 (right)).

Three referent breast cancer pathologists were willing to participate in the study, working with the same device and the same Android application. They manually and independently scored 3,756 digital biopsy patches.

## 3.2 Combination of scores from pathologists

The idea beyond this consensus process is to use a supervised learning method based on multiples experts that allow obtaining an estimated gold-standard that consensus labels assigned by experts and a mathematical model of each expert's experience based on the analyzed data and FISH results.

In this regard, the available FISH results will be used in two ways: (1) to generate a consensus model of expert opinions by training the machine learning method to obtain FISH-correlated results, and (2) to evaluate the performance of the machine learning method to obtain pathologists' consensus.

In addition, to evaluate the quality of the estimated gold-standard, area-under-curve (AUC) will be calculated using the estimated gold-standard versus labels according to the FISH results of each biopsy. Furthermore, to measure the reliability of the estimated gold-standard, AUC will be evaluated versus individual labels of each expert.

## 4. Preliminary Data Analysis

### 4.1 Dataset

The collected biopsies are related to 30 cases of invasive breast carcinomas with $3 - 4$ ROIS for each sample. Magnification of $20\times$ was used to crop $1834 \times 827$ non-overlapping rectangular patches, with $15 - 17$ patches for each ROI. The resulting dataset contains 1,252 biopsy patches with histopathological diagnoses (0, 1+, 2+, 3+). Figure 2 shows representative biopsy patches from each class. The manual scoring process was performed independently, per biopsy patch, by three referent experts with vast experience in HER2 breast cancer scoring.

### 4.2 Intra-expert variability

During the Android application design, it was considered to present the same biopsy patches to each pathologist in random order. In addition, the presentation of the same patches contemplates a previous flipping and 90-degree rotation transformation to increase the complexity of recognition. Thus, there are three scores associated with the same patch but in different presentations.

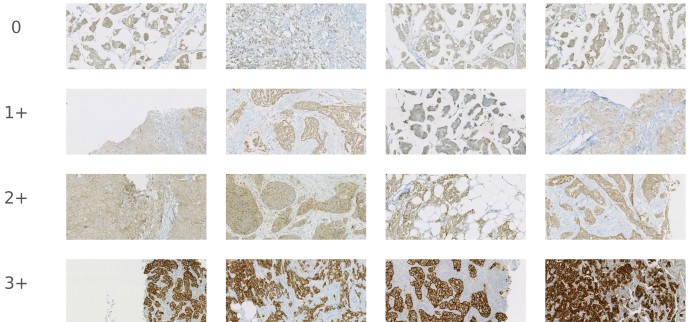

Figure 2: **HER2 breast cancer scoring dataset.** Representative images of 0, 1+, 2+ and 3+ HER2 breast cancer biopsy patches that showed total agreement among experts (Image size: $1834 \times 827$ pixels, $1\mu m \sim 2.27$ pixels).

To assess the robustness of each pathologist, Figure 3 shows intra-expert variability per class, according to each type of image (normal, rotated and flipped). It can be observed a clear consistency and robustness in the manual HER2 scoring for all pathologists. In this sense, the geometric transformations have no impact on the HER2 score and the distribution of scores remains similar regarding original or rotated or flipped images.

We used the Fleiss Kappa statistic (9) to measure the degree of intra-pathologist variability. In this sense, each presentation of the same biopsy section for manual scoring was considered a distinct entity. Regarding this, the three pathologists achieved a substantial intra-expert agreement with coefficients of 0.62, 0.67 and 0.64 for each expert, respectively ($\alpha = 0.05$).

### 4.3 Inter-expert variability

A critical aspect in the analysis of the dataset is the discussion of the inter-expert agreement distribution. As this dataset was built with the cooperation of three expert pathologists, there are three different agreement scenarios: only one expert (none agreement - NA), two experts (partial agreement - PA), or three experts agree on the same score for a given biopsy patch (total agreement - TA). The first set contains 211 biopsy patch scores, belonging to only 90 patches because a patch can be scored as three different scores by the three different pathologists. The second set contains 455 biopsy patches, meaning that 455 out of $1,252$ patches with a partial agreement and without overlapping. The third set contains 592 biopsy patches, with a total agreement between the three pathologists.

| Degree of agreement | 0 | 1+ | 2+ | 3+ | Number of patches |
|---|---|---|---|---|---|
| None agreement (NA) | 23 | 82 | 69 | 37 | 90 |
| Partial agreement (PA) | 33 | 195 | 181 | 46 | 455 |
| Total agreement (TA) | 14 | 110 | 55 | 413 | 592 |

Table 2: **Inter-expert agreement**

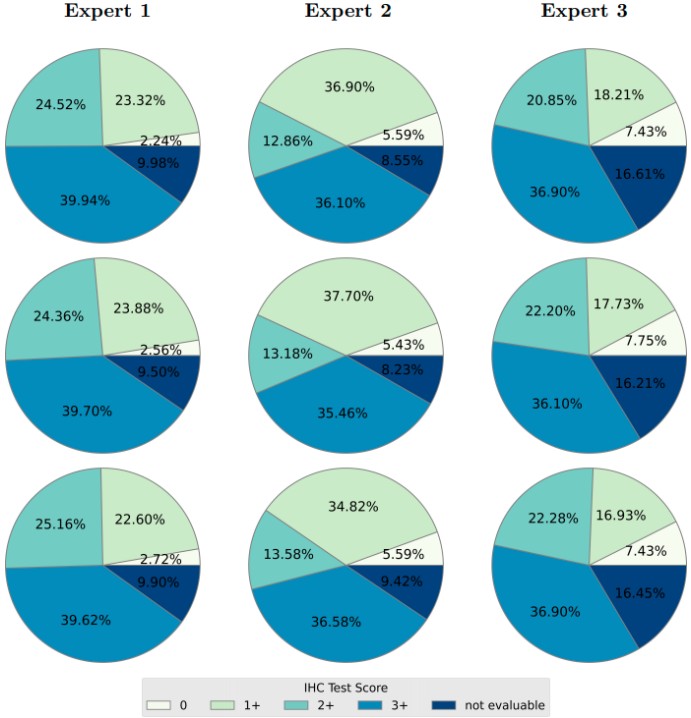

Figure 3: **Intra-expert variability per class.** Upper row shows intra-expert varibility for original images, middle row for rotated images, and last row for flipped images. The manual scoring shows a substantial intra-expert agreement with Fleiss' Kappa coefficient of 0.62, 0.67 and 0.64 for each expert, respectively ($\alpha = 0.05$).

Table 2 shows the number of biopsy patches per class for each agreement scenario. Considering the manual scoring agreement by one, two, or three expert pathologists, the score 0 was the least extensive set in all cases, incrementally decreasing assignment percentage as agreement among experts increases (see Figure 4). The score 3+ was the most significant set with total agreement of experts, almost 70%, but notably lower in partial agreement scenario (a bit up of 10%). The 1+ and 2+ scores are very similar in assignment percentage in scenarios without entire agreement among experts, with opposite behavior while increasing the agreement among experts. It is important to note that in the case of score 2+, less than 10% of the samples had a total agreement of the experts (only 55 biopsy patches).

The underlying complexity of the automatic HER2 scoring can be studied by evaluating the degree of agreement between different experts. Considering the number of patches that comprise the original dataset (without geometric transformations), Figure 5 shows the percentage of partial agreement and total agreement. There are some scores in which it was tough to reach an agreement, for example, score 2+. Only 14.5% of biopsy patches were classified as 2+ by two experts, and less than 4.5% reach a total agreement. It seemed to be the most challenging score from which to find agreement among experts. In contrast, for score 3+ is easier to reach total agreement among experts.

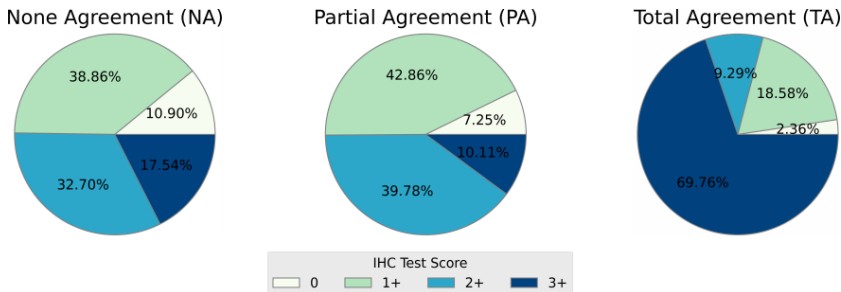

Figure 4: **Inter-expert degree of agreement.** (NA) Manual score by at least one expert assigning a score 1+ amounts to almost 39%. (PA) The score 1+ is the most extensive set again (almost 43%), while scores 0, 2+ and 3+ remain with similar distribution to the NA scenario. (TA) The score 3+ amounts to almost 70%, while 0 covers almost 2% and score 1+ doubles score 2+.

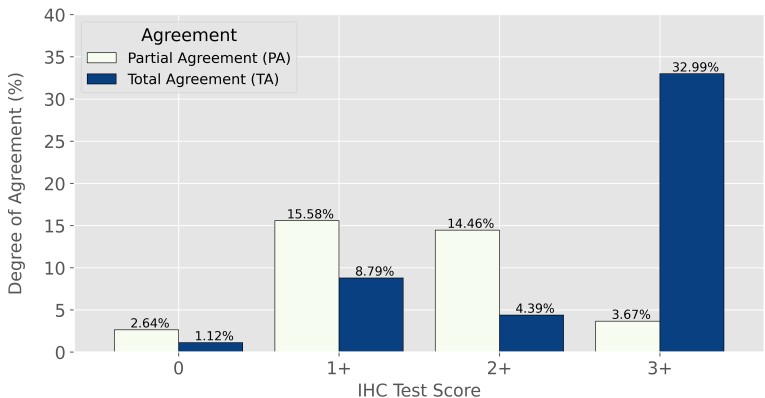

Figure 5: **Partial and total inter-expert agreement.** For each class, we show the percentage of partial and total agreement among experts normalized by the size of the total set of images.

To demonstrate the subjectivity of morphological analysis and dependence of the specialist who performs it, Figure 6 shows inter-expert variability per class. For scores 1+ and 2+, a high degree of variability was reached between two pathologists, whereas the discrepancy with the remaining expert was higher in the case of 1+. Scores 0 and 3+ showed a high degree of agreement among all experts. We calculated the Fleiss' Kappa coefficient (9) showing a moderate inter-expert agreement, with a coefficient of 0.55 ($\alpha = 0.05$).

## 5. Conclusions

To tackle the lack of a public breast cancer gold-standard, combining pathologists' opinions correlated with FISH results, in this article, we present the methodologies for collecting samples and expert opinions and assessing intra- and inter-expert variability. The collected

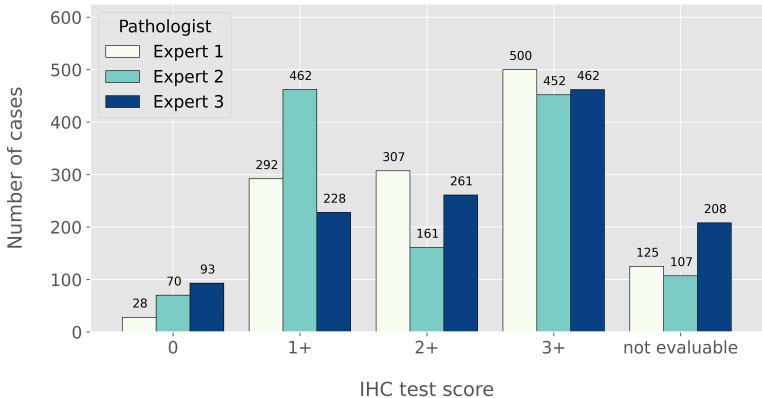

Figure 6: **Inter-expert variability in HER2 scoring.** We show the number of HER2 breast cancer biopsy patches that belong to each class according to each of the three experts. The expert manual classification shows a moderate agreement among experts with Fleiss' Kappa coefficient of 0.55 ($\alpha = 0.05$).

biopsies are related to 30 cases of invasive breast carcinomas. The resulting dataset contains $1,252$ biopsy patches that three referent experts manually and independently scored.

We used the Fleiss Kappa statistic to measure the degree of intra-pathologist variability, achieving a substantial intra-expert agreement with coefficients of 0.62, 0.67 and 0.64 for each expert, concluding that geometric transformations have no impact on the HER2 score. Concerning inter-expert variability, the number of patches with score 3+ significantly increases as agreement among experts increased. For score 2+, less than 10% of the biopsy patches had a total agreement of the experts. Fleiss' Kappa coefficient showed a moderate inter-expert agreement with 0.55.

This research aims to achieve, as the ultimate goal, the consensus opinion of various expert pathologists on HER2 breast cancer biopsy cases, using supervised learning methods based on multiple experts. In this sense, this paper suggests the main direction for future research: to use a supervised learning method based on multiples experts that allow obtaining an estimated gold-standard that consensus labels assigned by experts and a mathematical model of each expert's experience based on the analyzed data and FISH results. This gold-standard is for evaluating and comparing known techniques and future improvements to present approaches for automatic HER2 breast cancer biopsies.

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
