# OpenReview forum: "Towards a gold-standard of HER2 breast cancer biopsies using supervised learning based on multiple pathologist annotations"
_MICCAI.org/2021/Workshop/COMPAY — Reject_

### Official Review · Reviewer_yuPh · 2021-08-05
**??**

**Rating:** 6
**Confidence:** 4

**Review:**

The paper presents first steps towards a database of pathology images for HER2 diagnosis, with consensus gold standard by three experts. The final goal of the authors is a very interesting and useful one, but the work presented in the submission is rather preliminary. It describes a collection of data, how grading by three pathologists was obtained and intra- and interobserver variability. So far a standard approach to creating datasets. Obviously, good datasets with gold standards are needed and I applaud the efforts of the authors, but as the data does not seem to be available (yet?), the value of this submission is very limited. It mainly demonstrates observer variation for a particular application.
The paper mentions a consensus outcome and using deep learning somehow, but the details are not clear and the work is not included. A comparison to FISH is promised, but also not yet included.

The analysis of the observer results is valid and useful. The presentation of results is clear. Overall, the paper could be better structured. The  data seems to be presented twice (sections 3.1 and 4.1), the first paragraph of section 2 is superfluous, the mention of all the future plans without clear description is confusing, the number of references is somewhat high for a conference paper.

---

### Official Review · Reviewer_tL4T · 2021-08-22
**HER2 scoring demands an objective standard**

**Rating:** 4
**Confidence:** 4

**Review:**

Summary:

The authors assessed the intra- and inter-expert variability in HER2 scoring of breast cancer biopsies. The results confirmed the robustness of individual expert decisions on geometrically transformed images while signified the inter-expert disagreement on score 1+ and 2+ images.

 Pros of the paper:
-	The paper highlights the necessity of generating a gold standard for HER2 scoring in breast cancer biopsies.
-	Inter- and intra-expert variability is effectively evaluated for different scoring categories.

Cons of the paper:
-	In the conclusion the authors claim that “this paper suggests the main direction for future research: to use a supervised learning method based on multiples experts that allow obtaining an estimated gold-standard that consensus labels assigned by experts and a mathematical model of each expert’s experience based on the analyzed data and FISH results.” It is unclear how a model trained on variable ground truth can provide reliable predictions as to resolve the disagreement between experts.
-	The FISH-related analysis is not included in the result.

---

### Decision · Program_Chairs · 2021-08-25

Reject